# Membrane Cascade Type of «Continuous Membrane Column» for Power Plant Post-Combustion Carbon Dioxide Capture Part 1: Simulation of the Binary Gas Mixture Separation

**DOI:** 10.3390/membranes13030270

**Published:** 2023-02-24

**Authors:** Artem A. Atlaskin, Anton N. Petukhov, Anna N. Stepakova, Nikita S. Tsivkovsky, Sergey S. Kryuchkov, Kirill A. Smorodin, Irina S. Moiseenko, Maria E. Atlaskina, Sergey S. Suvorov, Ekaterina A. Stepanova, Ilya V. Vorotyntsev

**Affiliations:** 1Laboratory of Electronic Grade Substances Technologies, Mendeleev University of Chemical Technology of Russia, 125047 Moscow, Russia; 2Chemical Engineering Laboratory, National Research Lobachevsky State University of Nizhny Novgorod, 603022 Nizhny Novgorod, Russia

**Keywords:** flue gases, carbon dioxide, membrane, cascade, process design

## Abstract

The present paper deals with the complex study of CO_2_ capture from combined heat power plant flue gases using the efficient technological design of a membrane cascade type of «Continuous Membrane Column» for binary gas mixture separation. In contrast to well-known multi-step or multi-stage process designs, the cascade type of separation unit provides several advantages. Here, the separation process is implemented in it by creating two counter current flows. In one of them is depleted by the high-permeable component in a continuous mode, meanwhile the other one is enriched. Taking into account that the circulating flows rate overcomes the withdrawn one, there is a multiplicative increase in separation efficiency. A comprehensive study of CO_2_ capture using the membrane cascade type of «Continuous Membrane Column» includes the determination of the optimal membrane material characteristics, the sensitivity study of the process, and a feasibility evaluation. It was clearly demonstrated that the proposed process achieves efficient CO_2_ capture, which meets the modern requirements in terms of the CO_2_ content (≥95 mol.%), recovery rate (≥90%), and residual CO_2_ concentration (≤2 mol.%). Moreover, it was observed that it is possible to process CO_2_ with a purity of up to 99.8 mol.% at the same recovery rate. This enables the use of this specific process design in CO_2_ pretreatment operations for the production of high-purity carbon dioxide.

## 1. Introduction

The environment of our planet, at present, is continuously suffering from different anthropogenic impacts, which have dramatic effects, such as the atmospheric emission of pollutants, the pollution of soil and subsoil, the disposal of production and consumption waste, and the major one—global climate change. The main cause of this is the emission of greenhouse gases (water vapors, carbon dioxide, methane, etc.) generated by global industry. Among them, according to US Environmental Protection Agency, carbon dioxide is the most produced component, meanwhile 35% of it is produced by the transportation segment and 31% is emitted by coal power plants [1], as given in the circle diagrams in Figure 1.

Over the past century, the carbon dioxide concentration in the atmosphere has dramatically increased from 275 to 387 ppm, and this has already led to a tangible increase in the temperatures on the planet, namely, the average temperature of the Earth’s surface in the 21st Century exceeds the same value of the 20th Century by 0.8–1.2 °C. If the build-up of CO₂ continues at the current rates, by 2060, it will have passed 560 ppm, which is more than double the level of the pre-industrial times [2]. The developed climate models predict that the established trend will negatively affect the global climate by 2100 [3]. As its source is anthropogenic action, nowadays, the carbon capture and storage project (CCS) is a worldwide accepted strategy that is undertaking by about 50 operational facilities today. The International Energy Agency estimates that to limit warming to below 2 °C, 2.8 Gt (billion tons) of CO_2_ per year would need to be stored by 2050 [4]. Given that a current large-scale facility captures around one million tons per year, this would suggest that over 2000 facilities need to be being doing this in the next thirty years. 

The estimated CO_2_ emissions per kWh of a coal-based power plant is 958 g. Given the contribution of 31% to the overall emission of carbon dioxide made by the electricity industry, coal-powered plants are considered as one of the main emission sources. Nowadays, the amine scrubbing technique [5] dominates the acid gases capture industry and is conventionally applied in flue gas treatments, as well as in natural gas sweetening. Besides its indisputable technical capabilities, chemical absorption has also been regarded as more convenient for utilizing amine technology throughout the past few decades. However, despite the effectiveness of that approach at capturing low contents of CO_2_ in feedstock [6], there are a number of drawbacks [7], including the corrosion of pipelines and equipment, high investment costs, the loss of sorbent solution due to its degradation, as well as potential environmental threats. The most critical one is that the process has a high energy consumption requirement that is equal up to the 30% of power plant’s produced energy, resulting in a CO_2_ capture cost of 40–100 $/ton and rise in electricity price of 50–90% [8,9]. This forced the world’s chemical engineering society to design new energy efficient processes that agree with the modern green chemistry principles [10,11,12,13,14].

In that context, special attention should be paid to membrane-based processes, which are able to separate gaseous media under ambient conditions without phase transitions and do not require chemical reagents or a heat supply or its removal, resulting in energy efficient gas processing [15,16,17]. A shortlist of recent studies [18,19,20,21,22,23,24,25,26,27,28] presents promising results on the application of membrane-based techniques for carbon dioxide capture from flue gases. Thus, in [18], the authors perform a comparison between absorption, adsorption, and membrane separation, which resulted in the conclusion that the membrane technique appears to be the least energy consuming option. Favre et al. [19] considered fixed-site carrier membranes for CO_2_ capture and determined that wet CO_2_ capture application has advantages over solution-diffusion membranes. In [20], the authors proposed an efficient technique for the purification of a target component from low-permeable components. Another approach, which allowed the authors to increase the efficiency of membrane processes is an unsteady state technique with pulsed retentate withdrawal [21]. Vorotyntsev et al. presented a hybrid separation technique [22,23,24], which is called absorbing pervaporation, where the combination of a liquid absorbent and a membrane results in the high selectivity. In [25], the authors considered an industrial-scale spiral-wound membrane unit for CO_2_ capture. Bhattacharyya [26] proposed a hybrid membrane-solvent process for a post-combustion operation and claimed that it can offer great advantages and flexibilities for CO_2_ capture. In [27], the authors investigated a process that uses both N_2_- and CO_2_-selective membranes for capturing 90% of the CO_2_ from flue gases. The authors determined that the capture cost may be less than 30 $/ton CO_2_. Solten et al. [28] observed different cascade schemes for CO_2_ capture and determined the most promising solution based on two membrane stages with the recycling to feed.

A comprehensive study devoted to the simulation of the CO_2_ capture process using a two-step vacuum design and a two-step counter-flow/sweep design showed that the membrane-based technique produces 95% rich CO_2_, recovering 90% of it, and has a process cost of USD 23/ton [29]. A series of studies performed by Zhao et al. [25,26,27,28] presented a complex comparison of a number of membrane cascade designs (totaling 14 schemes) for CO_2_ capture from flue gases and energy consumption analysis, juxtaposing amine scrubbing. As a result, the two-stage design of a membrane cascade with a recycled retentate stream in the second stage was determined as being optimal due to it having the lowest energy consumption values during the production of 95% pure CO_2_ with 95% recovery.

In light of the abovementioned facts, the present work represents a comprehensive study of the CO_2_ capture process from a binary gas mixture (N_2_/CO_2_) using the membrane cascade type of «Continuous Membrane Column», including the determination of the optimal membrane material characteristics, the sensitivity study of the process, and a feasibility study. It was clearly demonstrated that the proposed process results in efficient CO_2_ capture, which meets the modern requirements in terms of the CO_2_ content (≥95 mol.%), recovery rate (≥90%), and residual CO_2_ concentration (≤2 mol.%). Moreover, it was observed that it is possible to process CO_2_ with a higher purity of up to 99.8 mol.% at the same recovery rate, which enables researchers to use this specific process design in CO_2_ pretreatment operations for the production of high-purity carbon dioxide.

## 2. Membrane Separation Unit Simulation

In order to perform a simulation study of the membrane gas separation process using the Aspen Plus environment (Bedford, MA, USA), a custom ACM user block was used. That block is an updated version of the hollow fiber membrane element, which was developed by Ajayi and Bhattacharyya during the DOE Carbon Capture Simulation Initiative (CCSI) [30]. This is a one-dimensional partial differential equation (PDE)-based multi-component, and it may be applied for materials in which permeation occurs according to the solution-diffusion mechanism. Here, gas permeances are independent of the pressures, concentrations, and stage cut. The separation process occurs under isothermal conditions. That model allows us to predict the value of the pressure drop along the fiber bore side and the shell side of a unit in accordance with the Hagen–Poiseuille equation for a compressible fluid. In this model, the gas mixture feeds the unit from the shell side of the hollow fibers and permeates to the fiber bore. The membrane module functions in countercurrent flows in a steady state mode. The model provides profiles of the component fluxes and concentrations, and the gas mixture behavior is assumed to be ideal. The equation-oriented structure enables the user to perform rating or design calculations depending on the variables being specified to satisfy the degrees of freedom.

## 3. Design of the Technological Scheme for CO_2_ Capture

The present study focuses on the CO_2_ capture process using the specially designed membrane cascade type of «Continuous Membrane Column». The principal scheme of that separation unit is given in Figure 2. During the separation process, the flue gases that need to be separated are continuously fed to the feed inlet placed in between the stripping and enrichment sections of the membrane cascade and are mixed with the gas flow leaving membrane unit 2 of the enrichment section. The permeate of membrane units 1 and 2 is evacuated using a vacuum pump and is sent further away (membrane unit 3) using a compressor. Due to this, the stripping section generates the prior concentrate of carbon dioxide on the permeate side of membrane unit 1 and processes the feed stream until the allowed CO_2_ level is reached in the residue flow. After the prior concentration of carbon dioxide has been reached, membrane unit 3 completes the last purification step, so that the desired level of 95 mol.% is reached, meanwhile membrane unit 2 saves the CO_2_ in the retentate of unit 3 enriching the permeate side of the cascade by carbon dioxide. Therefore, the configuration of the membrane separation device allows us to efficiently process the flue gases. The product stream, which is the permeate of membrane unit 3, contains no less than 95 mol.% of CO_2_. The residual stream, which is the retentate of membrane unit 1, contains no more than 2 mol.% of CO_2_. Additionally, the recovery rate is no less than 90%. A brief list of the simulation parameters is presented in Table 1. It is important to note that the scheme given in Figure 2 shows the basic elements of the cascade. Depending on the specific process parameters (feed stream pressure and/or temperature), it may be additionally upgraded with compressor units, heat exchangers, and other units, such as condensers, etc. 

## 4. Results and Discussion

Here, the calculation of the technological scheme of a membrane cascade type of «Continuous membrane column» for carbon dioxide capture from flue gases in combined heat and power plants was performed. A parametric analysis of the proposed scheme was carried out in order to determine the selectivity values of the membrane used and its area, providing the optimum ratio between the purity of captured CO_2_, the recovery rate, and the CO_2_ content in the residual stream. The goal of the process is to achieve a purity of the captured carbon dioxide of ≥95 mol.%, a recovery rate of ≥90%, and a concentration of CO_2_ in the residual stream of ≤2 mol.%. The flue gas parameters are listed in Table 1. It is important to note that all further calculations were performed at a feed mixture pressure of 0.15 MPa and a pressure in the permeate side of 0.02 MPa. These values were determined based on a number of literature sources addressing the issue of CO_2_ capture from CHP plant flue gases by membrane gas separation [29,31]. In addition, the pressure of 0.02 MPa is the minimum value that is achievable in practice [29], and given the pressure of the separated mixture at 0.1 MPa, compression to a value of 0.15 MPa seems to be economically justified.

Figure 3 shows the process flow diagram designed with the Aspen™ Plus flowsheet. Here, in addition to the three membrane units (one (M1) in the stripping section and two (M2 and M3) in the enrichment section), three compressors are used: C1, to compress the feed gas mixture flow to increase the partial pressure gradient of CO_2_, C2, to compress the captured CO_2_ to prepare it for storage or transportation, and VC (vacuum compressor), to evacuate the permeate side of the membrane units (M1 and M2) and further compress the permeated mixture prior to feeding membrane unit M3.

The key characteristics determining the appropriateness of a particular process design are the carbon dioxide content in the product and residual streams, and also, the recovery rate of carbon dioxide. Therefore, it is necessary to determine the influence of the process parameters of the proposed technological scheme and the ranges of the above-mentioned characteristics available for optimization. A parametric analysis of the proposed process flow diagram was therefore carried out.

### 4.1. Influence of Membrane Selectivity on CO_2_ Capture Efficiency

The membrane CO_2_/N_2_ selectivity value was determined for further calculations required to achieve the key process characteristics. Carbon dioxide permeability was set at 1000 GPU based on the parameters used in [29] and the commercial availability of membrane with a similar permeability—MTR Polaris™ (Newark, NJ, USA). The calculation was performed with the ultimate values of the considered membrane area range in the stripping and enrichment sections of 83,000 and 4500 m^2^, respectively.

Figure 4 shows the influence of membrane selectivity on the carbon dioxide content in the product and residual streams withdrawn from the enrichment and stripping sections of the membrane cascade, respectively. The graphs clearly demonstrate that the selectivity of the membrane has a significant influence on the process efficiency. Therefore, a membrane with a selectivity of lower than 12 cannot achieve a CO_2_ content of ≤2 mol.% in the residual stream, and moreover, a membrane’s selectivity being higher than 32 is necessary to produce CO_2_ with purity of 95 mol.% and higher. The calculation results are in good agreement with previous experiments, in which the deep purification of a low permeable component and CO_2_ capture was investigated [32,33]. With these experiments, it was shown that even when one is using a membrane with a low selectivity (2.5), it is possible to achieve a high-purity product (99.997 vol.%), which corresponds to a low content of a high-permeable impurity. In the case of the experimental study of the membrane cascade during CO_2_ capture, it was shown that the use of a membrane with a selectivity of eight did not allow us to achieve the required product purity, namely, in the limiting ratio of the withdrawn streams from the membrane cascade sections, a CO_2_ purity value of 91.23 vol.% was achieved. The obtained dependence of the withdrawn gas streams on the membrane cascade is explained by the fact that the separation of that mixture, even with a the low-selective membrane, results in the permeation, for most part, of carbon dioxide to the permeate side at a high stage cut value (>0.6). At the same time, the permeate of unit M1 was formed mainly by nitrogen, which in its turn did not allow us to achieve a sufficient CO_2_ content to create a high partial pressure drop in the enrichment section. As a result, the low driving force in the enrichment section prevents CO_2_ separation with the required product purity. Increasing the selectivity of the membrane (α(CO_2_/N_2_) ≥ 32) solves this issue. The results obtained, firstly, are in good agreement with the results presented in [29]; secondly, they demonstrate the possibility of using the gas transport characteristics of the MTR Polaris™ membrane for further calculations of the membrane cascade.

### 4.2. The Effect of the Membrane Area

Figure 5 demonstrates the influence of the membrane area in the stripping section on the carbon dioxide recovery rate. The graphs in the figure show that the membrane area has a significant effect on the carbon dioxide recovery rate. The graphs show the mutual influence of the stripping and enrichment section membrane areas on the characteristics of the process. Thus, using a membrane area of 4500 m^2^ in the enrichment section and a ~53,000 m^2^ membrane area in the stripping section are required to achieve the target CO_2_ recovery rate. At the same time, by reducing the size of the membrane area in the enrichment section, the required area in the stripping section area increases by 42.5% to ~75,400 m^2^. Furthermore, it can be seen that using the 1500 m^2^ membrane area in the enrichment section did not allow us to reach the required CO_2_ recovery rate in the considered range of the membrane area in the stripping section. The resulting dependencies are explained by the operating principle of the membrane cascade. The stripping section membrane area determines the permeate flow and the carbon dioxide content. The low partial pressure ratio across the membrane due to a relatively low CO_2_ content (17 mol.%) in the feed stream requires the process to occur at a high stage cut value in unit M1, which can be achieved only by usage of a large membrane area at a set feed pressure of 0.15 MPa. As the stream enriched with carbon dioxide (up to 63 mol.%) enters the enrichment section of the membrane cascade, a considerably smaller membrane area is required for its capture, and consequently, the separation can be performed at lower stage cut values. However, even a small reduction of the membrane area in the enrichment section (by 1000 m^2^) leads to the inefficient capture of CO_2_ and the return of a substantial amount of it to the stripping section of the cascade and an increase in the required membrane area. Based on the results obtained, it is reasonable to optimize the process flow diagram by changing the size of the membrane area in the enrichment section, as a relatively small increase in its area provides a significant reduction of the membrane material area in the stripping section.

In order to determine the range of membrane area values in the enrichment section available for membrane cascade optimization, the effect of the membrane area in this cascade section was determined at various fixed membrane area values in the stripping section. The results are shown in Figure 6. As can be seen from the graphs, and as noted earlier, increasing the membrane area in the enrichment section provides a significant increase in the CO_2_ recovery values, but this approach is only effective with a membrane area of more than 52,780 m^2^ in the stripping section. For smaller membrane areas in the stripping section, increasing the membrane area in the enrichment section does not help to achieve a ≥90% CO_2_ recovery rate. This is because the smaller membrane area in the stripping section does not ensure sufficient carbon dioxide enrichment of the stream entering the enrichment section. This, in turn, does not allow the creation of the necessary partial pressure ratio to recover more than 90% CO_2_ in the enrichment section of the membrane cascade even at the high values of the stage cut achieved by increasing the membrane area. Thus, in terms of the capital costs, further optimization is advisable with a membrane area of 52,780 m^2^ in the stripping section.

In order to verify the previously obtained results against another criteria, a residual carbon dioxide content in the stripping section retentate stream of ≤2 mol%, an analysis of the effect of the membrane area used on this factor was performed. Figure 7 shows the dependence of the carbon dioxide content in the residual stream of the membrane area in the stripping section. As in the previous case, it can be seen that both sections of the membrane cascade have an effect on whether or not the separation process target characteristic of ≤2 mol% CO_2_ in the residual stream is achieved. In contrast to the previously discussed relationship, here, the entire considered range of the membrane area in the enrichment section achieves the target value. Again, a small increase (across the entire membrane cascade) in the membrane area in the enrichment section allows the separation process to be performed with a significantly smaller membrane area in the stripping section, namely, implementing the process using a 4500 m^2^ membrane area in the enrichment section, and the required membrane area in the stripping section is ~53,000 m^2^, while reducing the enrichment section to 3000 m^2^ leads to an increase in the required area in the stripping section to 68,000 m^2^. The explanation for these dependencies boils down to a discussion of the earlier results. Here, the CO_2_ content of the residual stream from the stripping section is determined by the amount of CO_2_ that is withdrawn as a permeate in unit M1, i.e., the process stage cut in this unit and the ability to capture the majority of the carbon dioxide in the enrichment section. Thus, it was found that the previously established minimum membrane area in the stripping section fully meets the requirement for the residual CO_2_ content in the stripping section retentate stream. In addition, the combined results suggest that the minimum required membrane area in the enrichment section is 4500 m^2^.

The final step in determining the influence of the membrane area on the process characteristics was an analysis of the effect of this parameter on the purity of the captured carbon dioxide. The results, in the form of the dependence of the purity of the captured CO_2_ on the membrane area in the enrichment section, are presented in Figure 8. The graphs show that at all of the combinations of the membrane area sizes in the stripping and enrichment sections, the target purity of the product (≥95 mol.%) is achieved. Such dependencies are explained by the fact that all the values of the membrane area (from the considered range) allow us to concentrate the CO_2_ in the stripping section enough for the subsequent capture of this component, with more than 95 mol.% in the product stream. In addition, it was found that increasing the membrane area in the enrichment section leads to a decrease in the purity of the captured CO_2_. This is related to an increase in the value of the stage cut value in the M3 module. As the stage cut value increases, and taking into account a CO_2_ recovery rate of >90%, a small amount of nitrogen begins to permeate. On the other hand, as the membrane area in the enrichment section decreases, the purity of the captured CO_2_ increases significantly up to 99.82 mol.% for a membrane area of 75,400 m^2^ and up to 99.7 mol.% for a membrane area of 52,780 m^2^ (optimum value) in the stripping section. Such results suggest the possibility of high-grade CO_2_ capture. Taking into consideration the previously obtained results establishing values of the membrane areas in the stripping and enrichment sections of 52,780 and 4500 m^2^, respectively, based on the product recovery rate, it can be concluded that these parameters are optimal for the process of carbon dioxide capture from CHPP flue gases. Within the scope of the paper, the calculation was performed for hollow fiber membrane modules.

### 4.3. Feasibility Study for a Membrane Cascade Type of «Continuous Membrane Column» for Carbon Dioxide Capture from CHPP Flue Gases

As a result of the parametric analysis of the proposed technological scheme for the three-module membrane cascade configuration, the main technological parameters of the process were established (Table 2).

The following formula [29] was used to calculate the cost of CO_2_ extraction per ton: (1)$$C_{c}=\frac{\left(P\times T\times E\right)+\left(0.2\times C\right)}{F_{{\mathrm{CO}}_{2}}\times T}$$
where C_c_ is the cost of capture per ton of CO_2_, USD/ton CO_2_; P is the power required for CO_2_ capture equipment, kW; T is the CHPP capacity factor (operating time per year), h/year; E is the electricity cost, USD/kWh; C is the capital cost of equipment, USD; $F_{{\mathrm{CO}}_{2}}$ is the mass flow of captured CO_2_, ton/h.

The capital costs of the equipment can be calculated from the following simplified formula: (2)$$C=(A_{\mathrm{str}}+A_{\mathrm{enr}})\times S_{M}+S_{C1}+S_{C2}+S_{\mathrm{VC}}$$
where A_str_ and A_enr_ are the area of membrane used in the stripping and enrichment sections, respectively, m^2^; S_M_ is the cost of 1 m^2^ of membrane, USD/m^2^ (USD ~50/m^2^ based on the MTR Polaris™ membrane manufacturer [29]); S_C1_—cost of compressor unit C1, USD; S_C2_—cost of compressor unit C2, USD; S_VC_—cost of vacuum compressor, VC, USD.

The compression work was calculated according to the formula: (3)$$P=P_{C1}+P_{C2}+P_{\mathrm{VC}}$$
where P_C1_, P_C2_, and P_VC_ are the compression work of compressors C1, C2, and the vacuum compressor, respectively. P_C1_, P_C2_, and P_VC_ are each calculated separately according to the formula: (4)Pi=Lin×γγ−1×RTinnv×PoutPinγ−1γ−11000
where P_i_ is compression work, kW; L_in_ is compressor inlet flow, mol/s; γ is adiabatic expansion coefficient of the gas mixture; R is the universal gas constant; T_in_ is inlet gas temperature, K; n_v_ is compressor efficiency; P_out_ and P_in_ are compressor inlet and outlet pressures, respectively.

The adiabatic expansion coefficient of the gas mixture was calculated as follows: (5)$$\gamma =\frac{C_{p}^{{\mathrm{CO}}_{2}}{}_{p}\times y_{{\mathrm{CO}}_{2}}+C_{p}^{N_{2}}{}_{p}\times y_{N_{2}}}{C_{v}^{{\mathrm{CO}}_{2}}{}_{p}\times y_{{\mathrm{CO}}_{2}}+C_{v}^{N_{2}}{}_{p}\times y_{N_{2}}}$$
where C_p_ and C_v_ are the heat capacities of the pure components at constant pressure and temperature, respectively, J/(mole K); $y_{{\mathrm{CO}}_{2}}$ and $y_{N_{2}}$ are the molar fractions of CO_2_ and N_2_ in the inlet stream, respectively. The heat capacity values of the pure components were obtained from the Aspen™ Properties database.

The efficiencies of the vacuum and compression parts are generally dissimilar. Therefore, a formula has been applied to calculate the efficiency, establishing a correspondence between the efficiency and the pressure ratio at the inlet and outlet of the apparatus: (6)$$n_{v}=0.1058\times \ln\left(\frac{P_{\mathrm{in}}}{P_{\mathrm{out}}}\right)+0.8746$$

As a result of the calculation of the compression work, it was found that P_C1_ = 360, P_C2_ = 900, and P_VC_ = 1270 kW. Thus, the total compression work, P, is 2530 kW.

The cost of the compressor equipment providing a capacity of ~100 m^3^ min^−1^ varies over a fairly wide range, so it is reasonable to calculate the cost of this equipment by linking it to its capacity and assuming that 1 kW = USD 500 [29]. Hence, the costs of each compressor unit are: ~USD 178,000, USD 635,700, and USD 450,600 for C1, VC, and C2, respectively. Using a cost of 1 m^2^ of membrane (including housing costs) of USD 50, the capital cost is ~USD 4,129,350. Thus, assuming a CHPP capacity factor of 7446 h year^−1^ and an electricity cost of USD 0.04/kW, the cost of capturing a ton of CO_2_ is USD 31. Comparing with the two-step vacuum and two-step counter-flow/sweep designs, which were considered in [29], it is seen that the proposed cascade provides a lower capture cost than the two-step vacuum design does (USD 39 /ton CO_2_) and a higher cost than the counter-flow/sweep design does (USD 23 /ton CO_2_). Nevertheless, to perform an adequate and correct comparison, it is necessary to simulate the processes using the same model, with the same membrane mass transfer characteristics and the same feed properties. A future study will consider at least three technological schemes and four component feed mixture, including nitrogen, oxygen, water, and carbon dioxide.

## 5. Conclusions

As a result of the complex study devoted to the CO_2_ capture process from combined heat power plants using the membrane cascade type of «Continuous Membrane Column», it was found that this specific design produces an efficient separation. Namely, during the separation of the binary N_2_/CO_2_ mix with 17 mol.% of CO_2_, it is possible to capture more than 90% of the carbon dioxide with a purity of 97.2 mol.%, meanwhile the residual content is 1.87 mol.%. That separation occurs in a ~57,000 m^2^ membrane area, and the capture costs are USD 31 per ton. Taking into account the amine scrubbing CO_2_ capture cost range of USD 40–USD 100, the proposed membrane cascade design seems to be a better solution for CO_2_ capture. Moreover, it was observed that it is possible to process CO_2_ with a higher purity of up to 99.8 mol.% at the same recovery rate, which allows researchers to use this specific process design in CO_2_ pretreatment operations for the production of high-purity carbon dioxide. However, there are a number of membrane competitive designs that should be considered when one is choosing the appropriate technological scheme. In future studies, the most perspective designs will be simulated using same mathematical model for the adequate comparison of the processing costs.

## Figures and Tables

**Figure 1 membranes-13-00270-f001:**
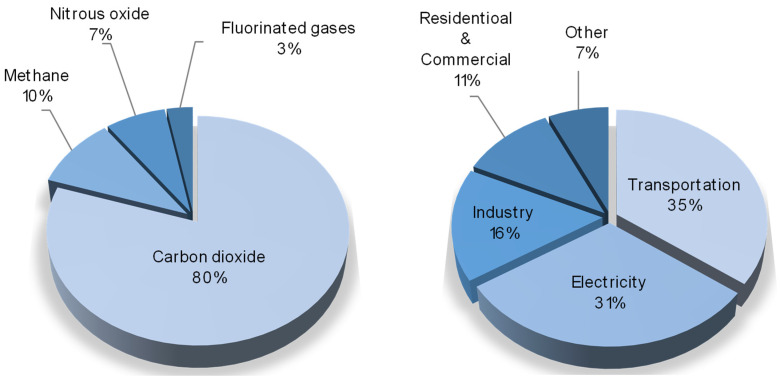
Some statistics on greenhouse gases emissions. **Left**—distribution of greenhouse gases by specific species; **right**—distribution of main greenhouse gases emitters.

**Figure 2 membranes-13-00270-f002:**
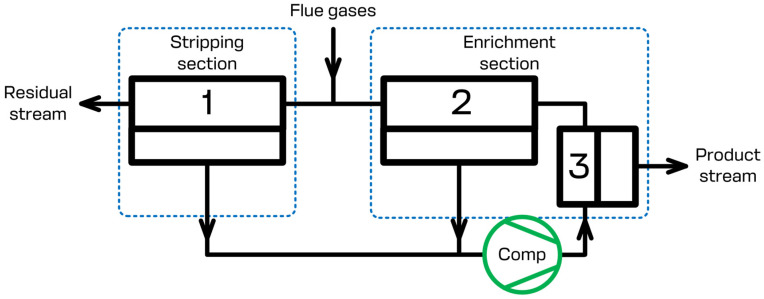
Principal scheme of the membrane column type of «Continuous Membrane Column». 1—membrane unit in the stripping section; 2, 3—membrane units in the enrichment section.

**Figure 3 membranes-13-00270-f003:**
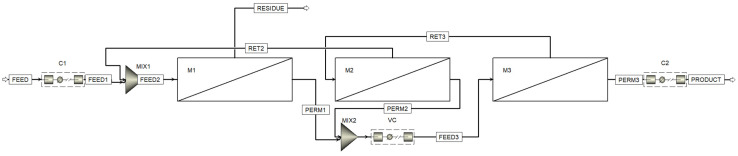
The process flow diagram designed in the Aspen™ Plus. M1, M2, and M3—membrane units; C1 and C2—compressors; VC—vacuum compressor.

**Figure 4 membranes-13-00270-f004:**
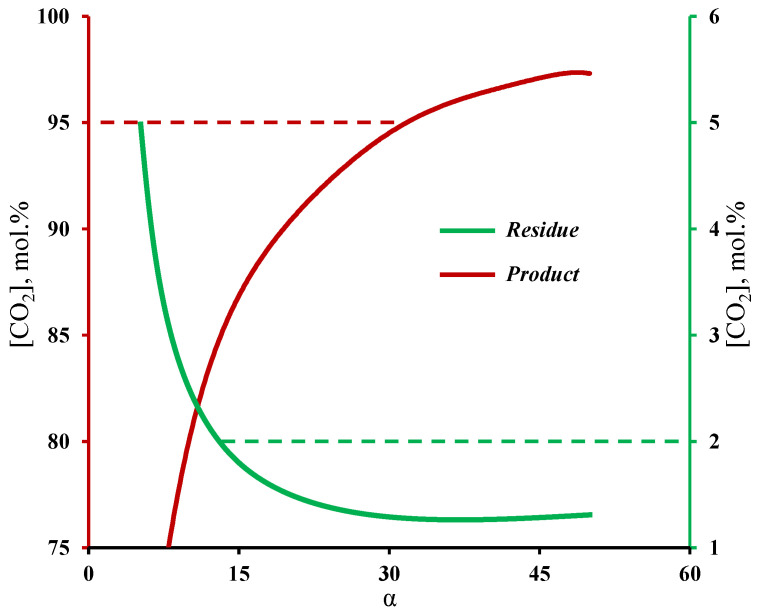
Dependence of carbon dioxide content in the streams withdrawn from the stripping and enrichment sections on the membrane selectivity at a fixed permeance value.

**Figure 5 membranes-13-00270-f005:**
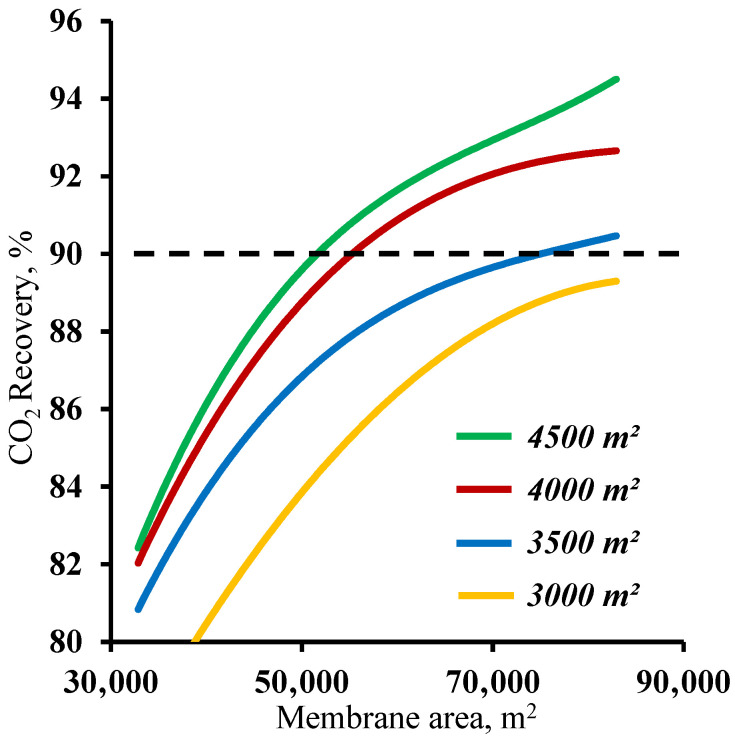
Dependence of carbon dioxide recovery rate on the membrane area in the stripping section (M1) for four different fixed membrane area values in the enrichment section (M2 and M3).

**Figure 6 membranes-13-00270-f006:**
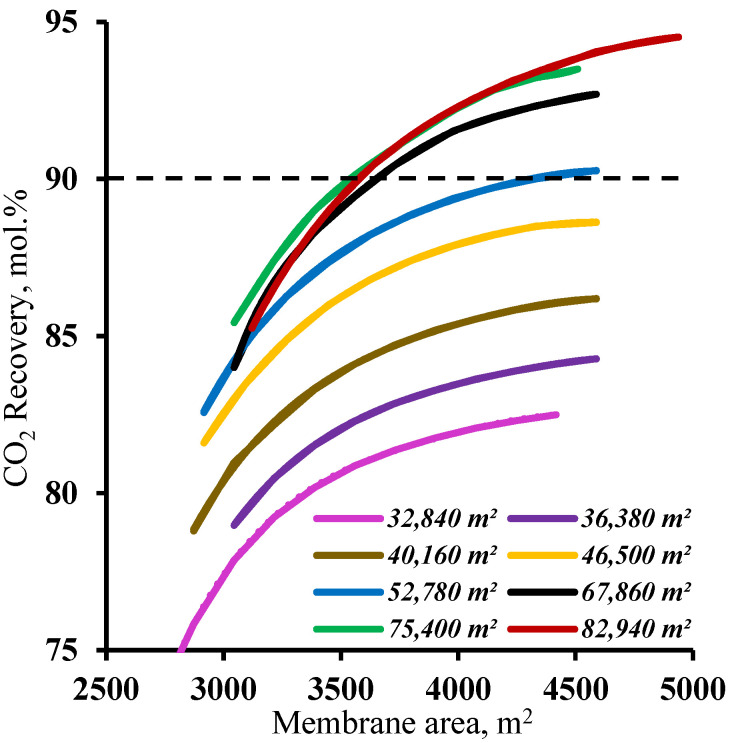
Dependence of carbon dioxide recovery rate on the membrane area in the enrichment section (M2 and M3) for eight different fixed membrane area values in the stripping section (M1).

**Figure 7 membranes-13-00270-f007:**
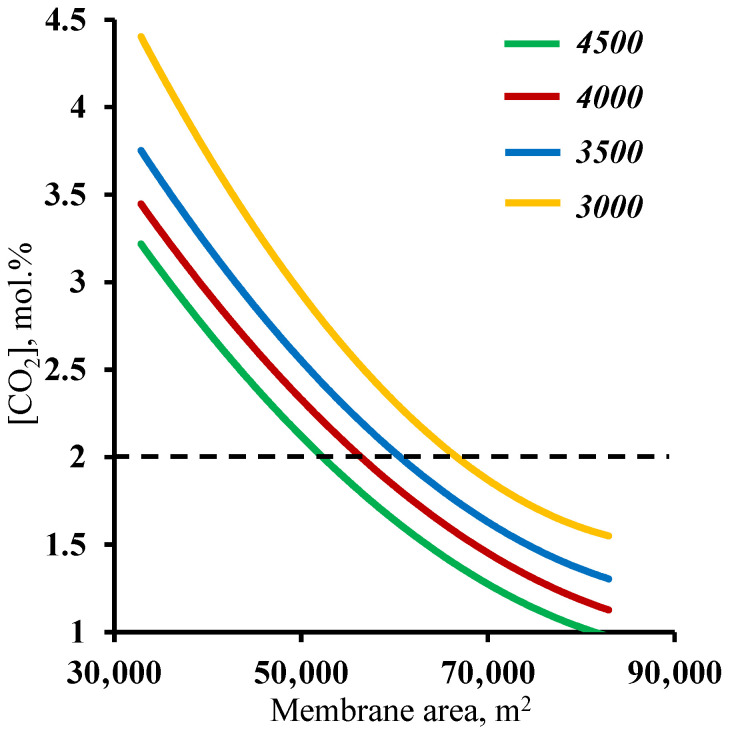
Dependence of carbon dioxide content in the residual stream on the membrane area in the stripping section (M1) for four different fixed values of the membrane area in the enrichment section (M2 and M3).

**Figure 8 membranes-13-00270-f008:**
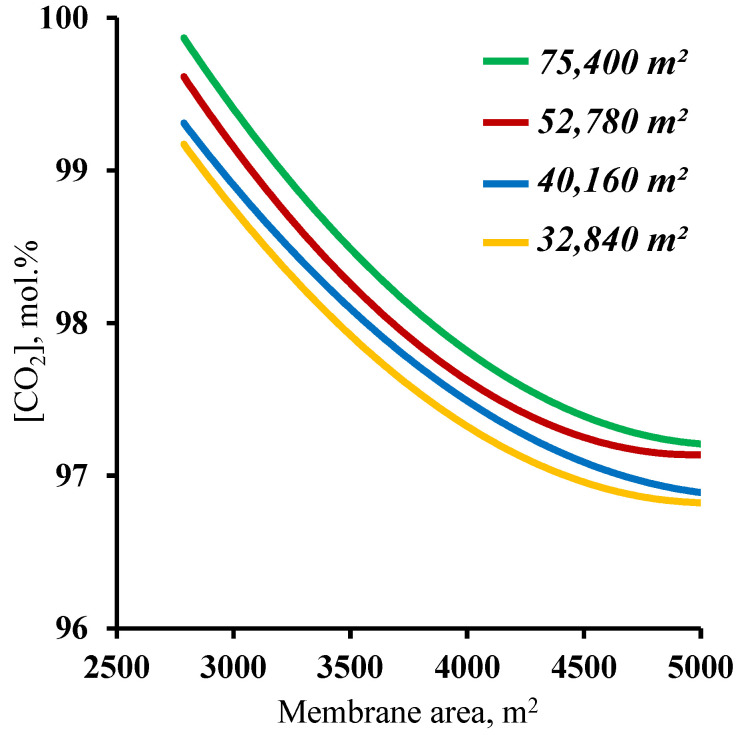
Dependence of carbon dioxide content in the product stream (withdrawn from the cascade enrichment section) on the membrane area in the enrichment section (M2 and M3) for four different fixed membrane area values in the stripping section.

**Table 1 membranes-13-00270-t001:** Input parameters of the separable gas stream.

Parameter	Value
Gas mixture inlet flow, kmol h^−1^	976,154
Pressure, MPa	0.1
Composition, mol.%	
N_2_	83
CO_2_	17

**Table 2 membranes-13-00270-t002:** Key process parameters of the CO_2_ processing in the membrane cascade during its capture from CHPP flue gases.

Parameter	Value	Units
Pressure in the feed side, MPa	0.15	MPa
Pressure in the permeate side, MPa	0.02	MPa
Membrane area, m^2^		
Stripping section	52,780	m^2^
Enrichment section	4500	m^2^
Membrane permeance, GPU	1000	GPU
Membrane selectivity for CO_2_/N_2_	50	
CO_2_ content, mol.%		
Product flow	97.2	mol.%
Residual flow	1.87	mol.%

## Data Availability

Not applicable.

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
