# Peer review of "Membrane Cascade Type of «Continuous Membrane Column» for Power Plant Post-Combustion Carbon Dioxide Capture Part 1: Simulation of the Binary Gas Mixture Separation"

_membranes, 2023, doi:10.3390/membranes13030270_

Round 1

Reviewer 1 Report

In the manuscript “Membrane Cascade Type of «Continuous Membrane Column» for Power Plant Post-Combustion Carbon Dioxide Capture Part  1. Simulation of the Binary Gas Mixture Separation“ the authors intend to propose an efficient process for CO2 capture. However, to publish this manuscript, high improvements should require.

1.       In the title appears “simulation of binary gas mixture separation” nevertheless binary does not write along the text (just once time in the conclusions).

2.       The state-of-the-art is outdated, 25/34 references are before 2019 with most of them in the period 2012-2014.

3.       The authors implement a model using the software aspen plus, (section 3.1), but there are no correctly detailed inputs (requirements and specifications in the flowsheet). 

4.       The model is not validated with any data (from the literature or experimental). In addition, the results reported here have lack of comparison with another process previously reported.

5.       The values of membrane size are ridiculous, 9,000 m2 is the size of a football stadium! If it is the real result, it is important a detailed description of the plant design.

6.       The section dedicated to the economic data is un-rigorous…there is a lack of data and it is not realistic, just 31$ per ton of CO2 captured. There is a high amount of data that is not considered. What is the expected durability of the membranes? The initial investment? This section should be highly improved.

In general, the idea is good and interesting but a lot of work it is still required to improve the quality of the manuscript, basically, new comparison data and rigorous information is highly recommended.

Author Response

The Authors want to express their appreciation to the Reviewer for the consideration of our article, thorough review, and pointing out the lack of the essential data.

In the manuscript “Membrane Cascade Type of «Continuous Membrane Column» for Power Plant Post-Combustion Carbon Dioxide Capture Part  1. Simulation of the Binary Gas Mixture Separation“ the authors intend to propose an efficient process for CO2 capture. However, to publish this manuscript, high improvements should require.

  1. In the title appears “simulation of binary gas mixture separation” nevertheless binary does not write along the text (just once time in the conclusions).

Authors comment:

Properties of the feed stream are given in the Table 1. Nevertheless, the Abstract and Introduction sections are updated with gas mixture composition details.

  1. The state-of-the-art is outdated, 25/34 references are before 2019 with most of them in the period 2012-2014.

Authors comment:

Authors agree with Reviewer’s comment. The references are updated.

  1. The authors implement a model using the software aspen plus, (section 3.1), but there are no correctly detailed inputs (requirements and specifications in the flowsheet).The model is not validated with any data (from the literature or experimental). In addition, the results reported here have lack of comparison with another process previously reported.

Authors comment:

Implemented model is a result of under the DOE Carbon Capture Simulation Initiative (CCSI). It is the open-source Aspen Plus user block, which is available on the GitHub. All data, assumptions and model structure are available by the link, which is given in References of the present study. Authors believe that it is incorrect to present the model as own development, because of that there is a brief description of that user model and the link to the source.

  1. The values of membrane size are ridiculous, 9,000 m2 is the size of a football stadium! If it is the real result, it is important a detailed description of the plant design.

Authors comment:

Total membrane area value is adequate and realistic, for example, the capture of CO2 from 600 MW power plant, which generates 500 m3/s of flue gases, requires from 1.3 to 3 million m2 (doi: 10.1016/j.memsci.2009.10.04). That study was performed by Richard W. Baker's collective, who are the MTR R&D specialists (Membrane Technology and Research https://www.mtrinc.com/). Moreover, the study deals with spiral wound membrane modules, which packing density is lower comparing to hollow fiber ones. Moreover, Enrico Drioli reports that the packing density of hollow fiber modules can reach up to 30 000 m2/m3 (Membrane Engineering for the Treatment of Gases Volume 1: Gas-separation Problems with Membranes, Chapter 5, Table 5.1). However, the more realistic value of packing density in commercially available modules with polysulfone fibers is near to 7 000 m2/m3. In the light of the above, the total membrane area of ~57 000 m2 requires less than 10 m3 for the membrane units.  

  1. The section dedicated to the economic data is un-rigorous…there is a lack of data and it is not realistic, just 31$ per ton of CO2 captured. There is a high amount of data that is not considered. What is the expected durability of the membranes? The initial investment? This section should be highly improved.

 Authors comment:

Unfortunately, authors are not agree with this Reviewer’s comment. The techno-economical evaluation performed including the capital costs (initial investment, which is required for separation unit), operating costs and plant operating time per year. That approach allows to compare the suggested technique with MTR solutions.

However, in order to correctly compare different technological scheme configurations, the simulation should be performed using the same aspen plus custom block. In further study, authors are going to simulate membrane cascade type of «Continuous Membrane Column», multistep and multistage designs (10.1016/j.memsci.2009.10.041) during the separation of N2/O2/H2O/CO2 gas mixture with feed flow of 500 m3/s.

Moreover, R.W. Baker demonstrates that capture cost per ton of CO2 using membrane method is in the range of 23 – 39 $ (10.1016/j.memsci.2009.10.041). Taking into account MTR experience and expertise over the CO2 capture process, authors tend to trust these results.

In general, the idea is good and interesting but a lot of work it is still required to improve the quality of the manuscript, basically, new comparison data and rigorous information is highly recommended.

Reviewer 2 Report

1.     Line 72, Page 2, authors should avoid lumped references. The authors should extract the key information from each reference.

2.     In section 3.2, the authors should support the findings with well-cited updated literature.

3.     The authors need to improve the quality of Figure 3. Currently, the Figure quality is not good.

4.     The authors should compare the results with the literature to highlight the significance of the article.  

5.     The list of references needs to be formatted especially references 1 and 4. Moreover, the authors need to put more updated references, i.e., 2020-2022.

Author Response

The Authors want to express their appreciation to the Reviewer for the consideration of our article, thorough review, and pointing out the lack of the essential data.

  1. Line 72, Page 2, authors should avoid lumped references. The authors should extract the key information from each reference.

Authors comment:

Authors agree with Reviewer’s comment. Each of the lumped references is described further in Introduction.

  1. In section 3.2, the authors should support the findings with well-cited updated literature.

Authors comment:

In that section of the present study authors discuss the behavior of the specific technological scheme configuration and search the ways for the optimization. In that way, it is not quite appropriate to support findings with results of principally different technological schemes because of the different dependencies (due to different configuration) and basics of membrane science (for example the recovery rate – product purity trade-off). Nevertheless, Section 3.3 is updated with the comparison of CO2 capture cost.

  1. The authors need to improve the quality of Figure 3. Currently, the Figure quality is not good.

Authors comment:

The quality of Figure 3 is improved. Image resolution is 2 368 x 503. Present quality of image allows to zoom in without quality loss.

  1. The authors should compare the results with the literature to highlight the significance of the article.  

Authors comment:

Authors believe that to make the most adequate and correct comparison between different technological schemes the simulation mast be performed using the same model and the same membrane properties. The Part 2 of that complex study will consider the comparison between the membrane cascade type of «Continuous Membrane Column», multistep and multistage designs (10.1016/j.memsci.2009.10.041) during the separation of N2/O2/H2O/CO2 gas mixture with feed flow of 500 m3/s. Nevertheless, Section 3.3 is updated with the comparison of CO2 capture cost.

  1. The list of references needs to be formatted especially references 1 and 4. Moreover, the authors need to put more updated references, i.e., 2020-2022.

Authors comment:

Authors agree with Reviewer’s comment. References are updated.

Reviewer 3 Report

In this manuscript, a membrane cascade type of «Continuous Membrane Column» was studied for CO2 capture from flue gas. The effects of membrane selectivity and membrane area on the residual CO2 concentration, CO2 recovery rate, and CO2 purity were investigated. The authors demonstrated that using the commercially available membranes with CO2 permeance of 1000 GPU and CO2/N2 selectivity of 50 (like MTR POLARIS membrane), the designed membrane cascade type of «Continuous Membrane Column» shows perspective for CO2 capture. Overall, this is a nice work offering a new membrane process design for CO2 capture and providing guidance for the practical application of membrane-based CO2 capture technologies. I would be happy to recommend publishing the paper in Membranes.

Here are some minor comments:

1. In Figure 3, the font size of the text is too small to read the figure easily. Please revise the figure to make the readers follow the figure more clearly. Also, it seems the C2 label is missing in Figure 3. Please double-check that.

2. In caption of Figure 8, M1 and M2 should be M2 and M3?

3. The authors have shown that the designed membrane process is competitive for CO2 capture with fixed CO2 concentration in flue gas (17%). It would be better if the authors could show how the CO2 concentration in flue gas affects the feasibility of this membrane cascade type of «Continuous Membrane Column» since for the power plant flue gas, the CO2 concentration can vary from 10% to 20%.

Author Response

The Authors want to express their appreciation to the Reviewer for the consideration of our article, thorough review, and pointing out the lack of the essential data.

In this manuscript, a membrane cascade type of «Continuous Membrane Column» was studied for CO2 capture from flue gas. The effects of membrane selectivity and membrane area on the residual CO2 concentration, CO2 recovery rate, and CO2 purity were investigated. The authors demonstrated that using the commercially available membranes with CO2 permeance of 1000 GPU and CO2/N2 selectivity of 50 (like MTR POLARIS membrane), the designed membrane cascade type of «Continuous Membrane Column» shows perspective for CO2 capture. Overall, this is a nice work offering a new membrane process design for CO2 capture and providing guidance for the practical application of membrane-based CO2 capture technologies. I would be happy to recommend publishing the paper in Membranes.

Here are some minor comments:

  1. In Figure 3, the font size of the text is too small to read the figure easily. Please revise the figure to make the readers follow the figure more clearly. Also, it seems the C2 label is missing in Figure 3. Please double-check that.

Authors comment:

The quality of Figure 3 is improved.

  1. In caption of Figure 8, M1 and M2 should be M2 and M3?

Authors comment:

The authors thank the referee for the typo found. Correction is done.

  1. The authors have shown that the designed membrane process is competitive for CO2 capture with fixed CO2 concentration in flue gas (17%). It would be better if the authors could show how the CO2 concentration in flue gas affects the feasibility of this membrane cascade type of «Continuous Membrane Column» since for the power plant flue gas, the CO2 concentration can vary from 10% to 20%.

Authors comment:

The authors completely agree with Reviewer. The present study deals with fixed CO2 concentration in order to determine the optimal membrane characteristics for the specific process and evaluate the range of membrane area, which is available for optimization on the example of binary gas mixture separation. Since the present paper is entitled «Part 1», authors are going to continue study and perform the simulation of the process during the separation of four component gas mixture (N2/O2/H2O/CO2), compare the membrane cascade with multistep and multistage configurations (10.1016/j.memsci.2009.10.041) and consider the effect of CO2 concentration in the feed.

Round 2

Reviewer 1 Report

The article could be accepted in the present form.